# Factor Structure of the Edinburgh Postnatal Depression Scale in a Sample of Postpartum Slovak Women

**DOI:** 10.3390/ijerph18126298

**Published:** 2021-06-10

**Authors:** Zuzana Škodová, Ľubica Bánovčinová, Eva Urbanová, Marián Grendár, Martina Bašková

**Affiliations:** 1Department of Midwifery, Jessenius Faculty of Medicine, Comenius University, 03601 Martin, Slovakia; zuzana.skodova@uniba.sk (Z.Š.); lubica.banovcinova@uniba.sk (Ľ.B.); eva.urbanova@uniba.sk (E.U.); 2Bioinformatic Unit, Biomedical Center Martin, Jessenius Faculty of Medicine, Comenius University, 03601 Martin, Slovakia; Marian.Grendar@uniba.sk

**Keywords:** Edinburgh Postnatal Depression Scale (EPDS), Slovakia, validity, postpartum depression

## Abstract

Background: Postpartum depression has a negative impact on quality of life. The aim of this study was to examine the factor structure and psychometric properties of the Slovak version of the Edinburgh Postnatal Depression Scale (EPDS). Methods: A paper and pencil version of the 10-item EPDS questionnaire was administered personally to 577 women at baseline during their stay in hospital on the second to fourth day postpartum (age, 30.6 ± 4.9 years; 73.5% vaginal births vs. 26.5% operative births; 59.4% primiparas). A total of 198 women participated in the online follow-up 6–8 weeks postpartum (questionnaire sent via e-mail). Results: The Slovak version of the EPDS had Cronbach’s coefficients of 0.84 and 0.88 at baseline (T1) and follow-up, respectively. The three-dimensional model of the scale offered good fit for both the baseline (χ^2^
_(df = 28)_ = 1339.38, *p* < 0.001; CFI = 0.99, RMSEA = 0.02, and TLI = 0.99) and follow-up (χ^2^
_(df = 45)_ = 908.06, *p* < 0.001, CFI = 0.93, RMSEA = 0.09, and TL = 0.90). A risk of major depression (EPDS score ≥ 13) was identified in 6.1% in T1 and 11.6% in the follow-up. Elevated levels of depression symptoms (EPDS score ≥ 10) were identified in 16.7% and 22.7% of the respondents at baseline and follow-up, respectively. Conclusions: The Slovak translation of the EPDS showed good consistency, convergent validity, and model characteristics. The routine use of EPDS can contribute to improving the quality of postnatal health care.

## 1. Introduction

Significant geographical and socio-demographical differences in the estimates of the prevalence of postpartum depression have been reported by several studies. In a systematic review by Shorey et al. [1], the average prevalence of depression was 17% among healthy mothers without a prior history of depression, with significant differences between geographical regions, with the Middle East having the highest prevalence (26%) and Europe having the lowest (8%). In a large-sample multinational study by Lupattelli et al. [2], significant differences in the prevalence of postpartum depression between European regions were found as well, with Eastern European countries having a significantly higher prevalence of postpartum depression. Similarly, differences in prevalence in high-income countries (9.5%) and low/middle-income countries (18.7%) were found in a systematic review by Woody et al. [3]; the overall pooled prevalence in this study was 11.9% among women during the perinatal period. The prevalence of depressive symptoms in a longitudinal ELSPAC study was 10–11% in a representative sample of postpartum women in the Czech Republic [4]. Only a few studies have estimated the postpartum depression occurrence in the Slovak Republic. A prevalence of 18% was reported in a small sample of Slovak postpartum women by Izakova [5]; similarly, 25% prevalence was reported by Banovcinova et al. [6]. 

In the etiopathogenesis of postnatal depression, genetic predisposition, together with anamnestic risk factors and the accumulation of psychosocial stressors, seems to play an important role. According to a systematic review by Hutchens and Kearney [7], the risk factors for postnatal depression are high life stress, lack of social support, current or past abuse, prenatal depression, marital or partner dissatisfaction, and prenatal depression. There is evidence that depressive symptoms that begin during the antenatal period tend to persist into the postnatal period. Up to one-third of women with postnatal depression have a diagnosis of depression or anxiety disorder in their medical history. Furthermore, as many as half of postpartum women with depression experienced depressive symptoms during pregnancy [8]. Depression or anxiety disorder in medical anamnesis prior to pregnancy is also a major risk factor for postpartum depression. However, a significant proportion of women experience depressive symptoms only during the postpartum period and have no increased risk of depression without a relationship to pregnancy or birth. A specific sensitivity of the mood-regulation system influencing pregnancy-related hormones might play a role in this case [9].

The association between the mode of delivery and postpartum depression remains unclear. Whereas some studies reported elevated levels of postpartum depressive symptoms amongst women who underwent operative birth or had perinatal factors such as significant blood loss or longer duration of stage II or III of labor [10]; other studies, including systematic reviews, found no significant association [11,12]. However, subjective perception of the birth experience and level of birth satisfaction might be more important risk factors of postpartum depression than the objective mode of delivery. Women’s negative perception of their birth experience, including factors such as lack of respect, privacy, support, inclusion in decision making, and feeling nurtured, may contribute to postnatal depression [12].

The Edinburgh Postnatal Depression Scale (EPDS) is a widely used measuring instrument for postpartum depression screening, providing a quick and simple administration and scoring system. The EPDS has been translated into over 60 languages and validated both as an antenatal and postpartum screen for minor or major depression in several countries [13]. High heterogeneity has been found regarding the sensitivity and specificity of the cut-off scores in different studies, possibly due to differences in study methodology, language, and diagnostic criteria used. Therefore, the validity of the EPDS as a screening instrument for postpartum depression may vary across different settings [14]. 

The EPDS was originally designed as a unidimensional measure. However, as shown in a comprehensive overview of the EPDS validation studies [15], most of the researchers confirmed the multidimensionality of the scale. In some studies, a two-factor structure has been found [16,17]; others have identified three factors of the EPDS: anxiety, anhedonia, and depression [18,19]. A theoretically driven four-factor model of the EPDS performed well in a Hungarian sample of postpartum women [20]. 

According to available information, the Slovak version of the EPDS has not yet been validated. The adaptation of the Slovak version of the Edinburgh Scale of Postnatal Depression and the examination of its psychometric properties and factor structure in a research sample of postpartum women in Slovakia were the main research aims in this study. 

## 2. Materials and Methods

### 2.1. Data Collection and Sample

A longitudinal follow-up design was used in this study, with data collection at two time points: T1 (paper and pencil questionnaire completed 2–4 days after birth) and T2 (electronic data collection 6–8 weeks postpartum). A convenient sampling method was used in the process of the data collection.

T1 data, baseline: The inclusion criteria at T1 were 2–4 days postpartum and informed consent. The exclusion criteria in T1 point were actual perinatal loss or stillbirth (previous perinatal loss in the anamnesis was not an exclusion criterium) and a history of severe psychiatric disorder in the anamnesis (psychotic disorder). Data for the T1 point were collected in 2 hospital birth centers in Slovakia: Bratislava (located in the capital city) and Martin (located in the central part of the country). Both centers are large university hospital facilities providing complex perinatal care, including specialized perinatological care for high-risk pregnancies and pathological births. Each participant filled out a paper and pencil questionnaire during their second to fourth day postpartum hospital stay (4 days is the standard length of hospital stay after a physiological birth in Slovakia). Data were collected by midwives working in the birth center between September 2018 and April 2020. Each participant in T1 data collection was personally approached by a midwife and invited to participate in the research. A total of 577 postpartum women participated in T1 data collection; the response rate for T1 was 82.3%.

T2 data, follow-up: All women who participated in the T1 data collection received an e-mail 6–8 weeks postpartum with the invitation to participate in the follow-up and an electronic version of the questionnaire (the e-mail addresses were provided when signing the informed consent letter). Altogether, 198 women participated in the follow-up (response rate of 34.9%). 

Most of the validation studies of the EPDS include women 6–8 weeks postpartum, or later. However, validation studies in Serbia, and Greece have also included women immediately after the birth [18,21]. Some of the authors [22,23,24] reported that the use of the EPDS in early postpartum is valid, and have argued that using the EPDS shortly after birth might have clinical value, especially in detecting the symptoms of anxiety or atypical depression, and in identifying women eligible for depression screening later during postpartum. Zanardo et al. [25] have found a strong association between high maternity blues scores and EPDS scores, and suggested that women who experienced strong symptoms of maternity blues may represent a distinct subgroup of postpartum women with a significantly increased risk of developing postpartum depression. Using the EPDS shortly after the birth also has significant practical advantages, as administering the questionnaire during the stay in the hospital allows approaching a larger proportion of postpartum women compared to online or telephone contact a few weeks of month postpartum.

Given the high attrition rate in T2 follow-up, differences between respondents and nonrespondents in the T2 follow-up were examined with chi-square statistics. The results of the nonresponse bias analysis showed significant differences regarding education (*p* ≤ 0.001) and type of birth (*p* ≤ 0.05). Among women who responded in the T2 follow-up, higher rates of participants with high education were found compared with the nonresponse group (68% vs. 53%). Women after physiological birth were also more likely to respond in T2 (79% in the response group compared with 72% in the nonresponse group). No significant differences between respondents and nonrespondents were found regarding age, parity, preterm birth, perinatal loss in anamnesis, complications in pregnancy, support person during birth, positive psychiatric anamnesis reported in T1, or positive depression symptoms (˂13 points in EPDS) in T1. Total EPDS scores in response and nonresponse groups were compared using Student independent samples *t*-test, and no significant differences were found (t = 1.90, *p* > 0.05).

### 2.2. Measuring Instruments

The Edinburgh Postnatal Depression Scale (EPDS) was used for both T1 and T2 of the data collection. The EPDS was developed by Cox et al. [26]. The EPDS is a 10-item self-rated questionnaire; each item asks about a common depressive symptom. Due to the specifics of the postpartum period, the EPDS does not contain items asking about somatic symptoms (as distinguishing somatic symptoms caused by physiological postpartum changes from those associated with depression is problematic). The scale also does not include items focused on assessing a mother–child relationship. Each EPDS item contains 4 response choices per statement (rated on a Likert scale). Items 3, 5, 6, 7, 8, 9, and 10 are reversely scored. The possible total score of the scale ranges from 0–30 points, with a higher score indicating a higher level of depression symptoms.

Different cut-off points of the EPDS have been used. According to the EPDS manual, second edition [15], a cut-off point of 10 or higher is recommended for research use, indicating elevated levels of depressive symptoms. For clinical use, a cut-off score of 12.5 has been shown to detect women at risk of major depression. As the EPDS is a screening measure, not a diagnostic tool, a woman who meets this threshold should be further assessed by a mental health professional for diagnosis. 

Permission to use the EPDS for this study was obtained from the Royal College of Psychiatrists (U.K.). The back-translation process of the EPDS was performed with an additional assessment of the accuracy of the translation. Strong emphasis was placed on possible ambiguous questions and culturally sensitive items. 

In a longitudinal follow-up (T2), the Zung self-rated depression scale was used along with the EPDS as a measure of the convergent validity of the EPDS. The Zung self-rated depression scale [27] is a self-reported 20-item scale measuring the symptoms of depression. The items’ responses are ranked from 1 to 4, with higher scores corresponding to elevated depressive symptoms. Ten positively worded items (2, 5, 11, 12, 14, 16, 17, 18, and 20) are scored reversely. The total score on the Zung depression scale indicates the level of depressive symptoms rather than a clinical diagnosis of major depression. The maximum total score is 80, and four categories can be identified based on the total score of the scale: 1, total score < 50 = no signs of depressive symptoms; 2, total score 50–60 = minimal signs of depression; 3, total score 60–69 = moderately to notably expressed signs of depression; and 4, total score > 70 = severe symptoms of depression. The Zung SDS is an established measure and in Slovakia, it is frequently used as a screening measure for depression. The Zung SDS has shown good psychometric properties and validity in the general population [28], in a population with depression [18], and in women of reproductive age [29].

At T1 baseline data collection, a sociodemographic and anamnestic questionnaire was used in the study together with the EPDS. The anamnestic questionnaire contained questions on basic sociodemographic variables (age and education) and perinatal data (type of birth, parity, support person during birth, preterm birth, perinatal loss in the anamnesis, and complications in pregnancy). Previous or actual onset of psychiatric disorders of the participants was measured by self-reported questions focused on a history of depression or other psychiatric illness before or during pregnancy.

### 2.3. Statistical Procedures

Statistical analysis was performed using IBM SPSS Statistics for Windows, version 25.0, and statistical software R, version 3.5.0 (2018). Student’s *t*-test for independent samples and ANOVA with Sheffe’s post hoc tests were used when testing differences in the mean EPDS sores in the different groups of respondents according to basic demographic and perinatal characteristics. For the analysis of the reliability of the EPDS, Cronbach’s α and the Spearman–Brown coefficient were employed. Chi-square tests were used for testing differences between respondents and nonrespondents in T2. The convergent validity of the EPDS was tested based on the correlation coefficients between the total EPDS score and the total Zung self-rated depression inventory score only in the T2 sample (6–8 weeks postpartum). Interclass correlation coefficients between the EPDS and Zung SDS were calculated as well; average measures value greater than 0.6 were considered acceptable.

Exploratory factor analysis (EFA) was employed to test the factor structure in both T1 and T2 samples. Multiple factor solutions (direct oblimin rotation) were run. Eigenvalues, scree plots, and the amount of variance explained were examined to determine the number of factors in each model. Factors with eigenvalues greater than one were retained, with a meaningful factor solution explaining at least 50% of the variance. An item loading significantly on a factor was determined by a loading of ≥0.3.

In the next step, confirmatory factor analysis (CFA) was performed in both the T1 and T2 samples with the goal of evaluating the EFA-based models’ fit to the data. The structural equation modeling (SEM) approach was employed to perform the CFA, using the statistical software R, version 3.5.0 [30], including the lavaan [31] and semPlot statistical packages [32]. A maximum likelihood (ML) approach to model estimation was adopted. Multiple goodness-of-fit tests were used to evaluate the models, including the chi-square statistic, comparative fit index (CFI), toot mean square error of approximation (RMSEA), and the Tucker–Lewis index (TLI). A CFI greater than 0.90 was employed as an indicator of an acceptable fit to the data, and a CFI equal to or greater than 0.95 indicated a good fit to the data. An RMSEA less than 0.08 was a threshold for an acceptable fit to the data; values of less than 0.05 indicated a good fit to the data. TLI values greater than 0.9 were considered the threshold for a good model fit [33,34].

### 2.4. Ethical Issues

All the participants were thoroughly informed about the project aims and ethical issues (anonymity, personal data protection, and voluntary participation). An informed consent letter was signed by each participant prior to their participation. Ethical approval was obtained from the Ethical committee of the Jessenius Faculty of Medicine in Martin, Slovakia, no. EK 36/2018.

## 3. Results

### 3.1. Distribution of the EPDS Scores

The basic demographic and anamnestic characteristics of the respondents who participated at both T1 and T2 are provided in Table 1, along with the mean EPDS scores in the different groups of participants. The T1 sample included 577 postpartum women with a mean age of 30.6 ± 4.9 years. Altogether, 375 (65%) participants were from Martin university hospital and 202 (35%) from Bratislava university hospital. Most of the respondents had higher education (57.9%), and 59.4% were primiparas. The majority of women in the research sample (73.5%) had a vaginal birth.

There were 198 postpartum women participants at T2, with a mean age of 30.9 ± 4.8 years; most of them had higher education (68.2%). Altogether, 58.1% of women were primiparas, and 79% had a vaginal birth. More detailed information on the study participants at T1 and T2 is provided in Table 1. In the T1 sample, statistically significant differences in EPDS scores were found regarding parity (primiparas scored significantly higher in the EPDS than multiparas), type of birth (women after operative birth scored significantly higher in EPDS than women after spontaneous birth). In both T1 and T2 samples, significant differences regarding the history of psychiatric disorders were found: women with a history of psychiatric disorders scored significantly higher than those without a history of psychiatric illness).

The mean EPDS score T1 at (2–4 days postpartum) was 5.24 (SD = 4.35), and 6.21 (SD = 4.93) at T2 (6–8 weeks postpartum); the difference in the EPDS scores between samples was statistically significant (t = −2.61, *p* ≤ 0.009).

A risk of major depression (EPDS score ≥ 13) was identified in 6.1% of women at T1 and 11.6% women at T2. An elevated level of depression symptoms (EPDS score ≥ 10) was reported by 16.7% and 22.7% of the respondents at T1 and T2, respectively. According to the results of the Zung self-rated depression scale, 17.7% of women reported having mild signs of depression, and 5.1% as having moderate signs of depression (results only from T2 sample). 

The mean EPDS score at T1 (2–4 days postpartum) was 5.24 (SD = 4.35), and 5.71 (SD = 4.42) at T2 (6–8 weeks postpartum); the difference in the EPDS scores between samples was not statistically significant (t = 1.288, *p* ≤ 0.19).

The risk of major depression (EPDS score ≥ 13) was identified in 6.1% of women at T1 and 8.4% women at T2. An elevated level of depression symptoms (EPDS score ≥ 10) was recorded in 16.7% and 19.5% of respondents at T1 and T2, respectively. According to the results of the Zung self-rated depression scale, 16.8% of women were identified as having mild signs of depression and 2.6%, as having moderate signs of depression (results only from the T2 sample). 

### 3.2. Convergent Validity

The total score of the Slovak version of the EPDS was statistically significantly correlated with the Zung self-rated depression scale scores (Pearson’s r = 0.718, *p* ≤ 0.001) in the sample of 204 postpartum women (T2, 6–8 weeks postpartum). According to Cohen’s interpretation of Pearson correlation coefficients, this represents a strong correlation between variables. The interclass correlation coefficients calculation showed an average measure value of 0.798 (95% CI: 0.73, 0.85). These results showed that the construct validity of the Slovak version of the EPDS is satisfactory.

### 3.3. Reliability of the EPDS Scores

Table 2 shows the internal consistency of the individual as well as total EPDS scores. The EPDS showed good reliability (Cronbach’s α 0.84 at T1 and 0.88 at T2). The α coefficients for individual EPDS items were above 0.80, indicating good homogeneity. The Spearman–Brown coefficient was 0.80 at T1 and 0.87 at T2, indicating the good split-half reliability of the scale.

### 3.4. Exploratory Factor Analysis (EFA) of the Data from the T1 Sample 

The Kaiser–Meyer–Olkin measure of sampling adequacy for T1 data was 0.82, and the result of the Bartlett’s test of sphericity was significant (*p*-value < 0.001), which indicated that the collected data were suitable for factor analysis. 

The results of the EFA revealed two components with eigenvalues greater than one: the first was 5.605, and represented a factor consisting of items 3–10; the second was 1.333 and contained items 1 and 2. However, there were items that loaded significantly on both factors: item 6 loaded almost similarly on factors 1 and 2, and item 8 loaded higher on factor 1 and lower but significantly on factor 2 (Table 3). The combination of the two factors explained 57.5% of the variance, and the correlation between factors was 0.43. the internal reliability of the first factor was 0.90, and 0.86 for the second factor.

The three-factor solution was suggested by parallel analysis. This solution contained a first factor (items 7–10); a second factor, which comprised items 1 and 2; and a third factor with items 3–5. Item 6 loaded significantly on both factors 1 and 2, with higher loading on factor 2. Item 10 loaded significantly on all three factors, with the highest loading on factor 1. The combination of the three factors explained 58.7% of the variance, and the correlation between factors varied from 0.30 to 0.64. the internal reliability of the first, second, and third factors was 0.87, 0.86, and 0.84, respectively. 

### 3.5. EFA of the Data from the T2 Sample 

The Kaiser–Meyer–Olkin measure of sampling adequacy for T2 was 0.82, and the *p*-value for Bartlett’s test of sphericity was less than 0.001. 

The results of the EFA revealed only one component with an eigenvalue greater than one (eigenvalue = 6.52). The factor loadings for each item ranged from 0.66 to 0.88 (Table 4), and this single factor explained 61.6% of the variance. Cronbach’s α of the factor was 0.94.

The two-factor solution was suggested by parallel analysis. This solution contained a first factor (items 1–2, 6, and 7–10) and a second factor that comprised items 3–5. Items 3 and 6 loaded significantly on both factors 1 and 2. The combination of the two factors explained 58.4% of the variance, and the correlation between factors was 0.76. The internal reliability of the first and second factors was 0.93 and 0.87, respectively.

The three-factor solution for the T2 data was examined as well. This solution contained a first factor (items 1 and 7–10); a second factor, which comprised items 2 and 6; and a third factor containing items 3–5. Item 1 loaded significantly on both factors 1 and 2, with a higher loading on factor 1, and item 6 loaded significantly on the second and third factors, with a higher loading on factor 2. Item 8 also loaded significantly on factors 1 and 2, with a slightly higher loading on factor 1 (Table 4). The combination of the three factors explained 56.1% of the variance, and the correlation between factors was 0.95–0.97. This three-factor model differs from the three-factor model identified by EFA using T1 data in one feature: in the T2 EFA model, item 1 loaded significantly on the first factor together with items 7–10; in the T1 EFA model, item 1 loaded on the second factor together with items 2 and 6. 

### 3.6. Confirmatory Factor Analysis (CFA)

The first step in the CFA was the evaluation of the two multidimensional structural models (two-factor and three-factor solutions) as well as the unidimensional model on data from the T1 baseline sample. As shown in Figure 1, the structure of the three-factor model of the Slovak version of the EPDS was found to have a good fit with the baseline data (χ^2^_(df = 32)_ = 159.84, *p* < 0.001; CFI = 0.94, RMSEA = 0.08, and TLI = 0.91). The two-factor solution was found to poorly fit the baseline data; the CFA and RMSEA values did not meet the threshold for an acceptable fit to the data (χ^2^_(df = 45)_ = 1982.96, *p* < 0.001; CFI = 0.87, RMSEA = 0.114, TLI = 0.82). Similar results were found for the unidimensional model of the EPDS (χ^2^_(df = 45)_ = 1982.78, *p* < 0.001; CFI = 0.80, RMSEA = 0.138, TLI = 0.74).

As the second step in CFA, both multidimensional structural models (two-factor and three-factor solutions) and the unidimensional model on data from the T2 sample were evaluated. A single-factor model of the Slovak version of the EPDS was found to poorly fit the data (χ^2^_(df = 45)_ = 908.06, *p* < 0.001, CFI = 0.88, RMSEA = 0.12, TLI = 0.84). The two-factor model showed an acceptable fit to the T2 data (χ^2^_(df = 45)_ = 908.06, *p* < 0.001, CFI = 0.93, RMSEA = 0.09 and TLI = 0.90), as did the three-factor EPDs model (χ^2^_(df = 45)_ = 908.06, *p* < 0.001, CFI = 0.93, RMSEA = 0.09 and TLI = 0.90).

An additional confirmatory factor analysis of the T2 data was performed as well using the three-factor model identified on T1 data. This step of the CFA revealed similar results: an acceptable fit of this model to the T2 data (χ^2^_(df = 45)_ = 908.05, *p* < 0.001, CFI = 0.94, RMSEA = 0.09, and TL =0.91) (Figure 2).

## 4. Discussion

To the best of our knowledge, this is the first study examining the factor structure and psychometric properties of the Edinburgh Postnatal Depression Scale in Slovakia. The Slovak translation of the EPDS showed good consistency, convergent validity, and acceptable model characteristics among postpartum women. The three-factor model best fit the Slovak data for both the baseline data (2–4 days postpartum) and the follow-up data (6 weeks postpartum). The three dimensions identified by factor analysis in our study might be seen as representing three factors: (1) depression (items 7–10 with/without item 1), (2) anxiety (items 3, 4, and 5), and (3) anhedonia (items 2 and 6, with/without item 1). Despite the authors of the EPDS presuming its single-dimensional character [26], many studies have reported the multidimensional nature of the EPDS, as shown in a study by Kozinszkyi et al. [20]. Our study findings agree with previous validation studies of the EPDS and confirm the multidimensionality of the scale. The three-factor model in the present study is similar to those found by Coates et al. [19] on a large representative U.K. sample and Odalovic et al. [18] on a smaller Serbian sample. The two-factor model was identified as an alternative model with an acceptable fit for the follow-up data (6 weeks postpartum) in our sample: (1) items 1, 2, and 7–10; (2) items 3, 4, and 5. This two-factor model is identical to the two-factor model proposed by Gollan et al. [17] in a study on a large representative USA sample. However, the two-factor model performed unsatisfactorily for our T1 data, which led us to the conclusion that the three-factor model best fit the Slovak data.

Both exploratory and confirmatory factor analysis revealed differences in the models for the T1 and T2 samples in our study. A possible explanation for these discrepancies might be the differences between the T1 and T2 samples in our study. The high attrition rate in the T2 sample, which resulted in a low number of participants at T2, might have contributed to the differences in the results of the EPDS factor analysis of the T1 and T2 samples. Another factor potentially influencing the different results of EFA and CFA in the T2 sample is the nonresponse bias according to lower education and operative birth in our sample. Moreover, the validity of using the EPDS shortly after birth might be problematic. Although the use of the EPDS early postpartum was found to be valid by some studies [24], a critical point against the EPDS validity 1–2 weeks postpartum was recently raised [35]. Petrozzi et al. [22] argued that symptoms measured by the EPDS in a short time postpartum are symptoms of anxiety rather than depressive symptoms, and they propose more focus on differentiating the EPDS anxiety and depression subscales. Dennis et al. [24] showed that the 1-week EPDS accurately classified approximately half of women at 16 weeks postpartum with elevated EPDS scores. Thus, it seems that using the EPDS shortly after the birth needs to reflect the heterogeneity in the EPDS results at different time points, and the relevance of using the EPDS in early postpartum needs to be examined closer. 

In our sample, the risk of major depression (EPDS score higher than or equal to 13 points) was identified among 6.1% of women at T1 and 11.6% at the 6–8 weeks follow-up. When using the EPDS cut-off point of 10 points or higher, the prevalence of depressive symptoms was 16.7% at T1 and 22.7% at T2. Lupattelli et al. [2] reported a 30% prevalence of mild to moderate depressive symptoms (EPDS scores 10–16), and a 5–6% prevalence of moderate to severe symptoms (EPDS scores > 17) in Eastern European countries, which is higher than in our study. Similarly, higher prevalence compared with our study (24.8%) was found among Serbian postpartum women using the 13-point criterion [18]. One of the possible reasons for this might be that data were collected over a longer timespan after birth (1 year) in both of these studies. However, depressive syndrome prevalence in our study was similar to the findings of an ELSPAC study in a large representative sample of Czech women 6 weeks postpartum, where the prevalence of 21.9% was found (using the 10-point criterion), and 11.8% when the stricter cut-off point was used [4]. Similar to our study, 10,1% prevalence of 6–8 weeks postpartum was found in a study by Coates et al. [19] in a large UK sample; and 10.8% prevalence of 3–24 weeks postpartum was found by Nagy et al. [36] in Hungarian sample. In both studies, a 13-point cut-off of the EPDS criterion was used. In a review by Lyubenova et al. [37], the pooled prevalence of postpartum depression ranged from 27.8% when a cut-off of nine points was used to 9.0% when a 14-points cut-off was employed. Comparison of depressive symptoms prevalence across different studies is complicated using different cut-off points and data collection time spans; however, it seems that the prevalence of depressive symptoms found in our study is similar to that of other studies, and depressive symptoms occurrence might have an increasing tendency with time during the postpartum period. The prevalence of depressive symptoms among postpartum women across research studies highlights the importance of the detection of depression as a part of postpartum screening programs. According to Cox [38], in some countries, the EPDS is included in the national screening programs for perinatal women (for instance, in the USA, Sweden, and Australia), and in the U.K., its use is recommended).

Some of the methodological limitations of this study need to be mentioned. When comparing our sample with birthing women in Slovakia based on national statistical reports [39], some of the sociodemographic and perinatal characteristics in our sample differed significantly from the population of Slovak women giving birth. The respondents in our sample were more often highly educated (58.2%) compared with the population of birthing women in Slovakia (34.7%) and more often primiparas (59.9% vs. 45.9% in the whole population). The type of birth in our sample was vaginal in 73.8%, whereas it was 67.8% in the whole population of women giving birth in Slovakia. These differences influence the possibility of generalizing our results to the whole population of Slovak women. The second limitation of the study is the high attrition rate at T2, which may have occurred due to the differences in the data collection process: women were approached personally during baseline T1 data collection, which contributed to the higher response rate compared with online data collection via e-mail for the T2 follow-up. The analysis of response bias showed that women with lower education and after operative birth were less likely to respond at T2. As both these characteristics might be seen as risk factors for developing postpartum depression, these differences might have influenced the results of our analysis. Using the Zung depression scale in the process of validation is also a possible methodological limitation of our study, as well as convergent validity only being assessed for the T2 sample. Although the Zung SDS has been established as a widely used screening measure and has been validated in the general population [28], a population with depression [40], and women of reproductive age [29], the appropriateness of its use among perinatal women has not been sufficiently explored. No history of depression diagnosis (other than one self-report question) is also a methodological weakness in our research. Another limitation in our study is that we did not analyze the possible impact of ethnicity on our results. The largest ethnic minority in Slovakia, Roma, accounts for approximately 8–9% of the population. A significant proportion of this ethnic minority lives in socially excluded communities with severely disadvantaged socioeconomic conditions, which can significantly impact postpartum depression prevalence in this group. Finally, we have considered only cut-off values validated in different countries in our study, and we did not establish the exact cut-off values appropriate and sensitive for use in the Slovak population, as we did not consider a formal diagnosis of postnatal depression in our research. This issue needs to be addressed in future studies, which should focus on establishing the cut-off scores sensitive to the characteristics of the Slovak population of postpartum women, using, for instance, ROC curves.

## 5. Conclusions

The Slovak translation of the EPDS showed good consistency, convergent validity, and good model characteristics in a sample of postpartum Slovak women. Postpartum depression remains undiagnosed and untreated in a high percentage of cases in Slovakia. Multidisciplinary cooperation of experts, particularly in the fields of psychiatry, clinical psychology, gynecology, and midwifery, is required to improve postpartum depression screening and treatment. Routine screening for postpartum depression may significantly help to identify women with an increased risk of developing depression, thereby contributing to improving disease prevention and effective treatment. The administration and scoring of the Edinburgh Postnatal Depression Scale are quick and simple, and its use in routine screening programs in postpartum care may contribute to improved quality of health care with increased emphasis on mental health in Slovakia. Further studies should focus on establishing the sensitivity and specificity of the Slovak version of the EPDS on a larger and more representative sample using a psychiatric diagnostics interview as the acknowledged standard in the EPDS validation. 

## Figures and Tables

**Figure 1 ijerph-18-06298-f001:**
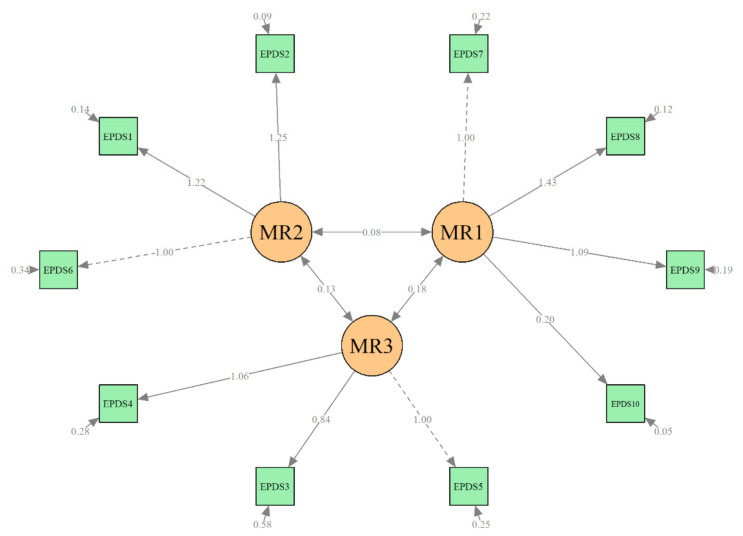
Confirmatory factor analysis of T1 data, 3-factor model (3–4 days postpartum).

**Figure 2 ijerph-18-06298-f002:**
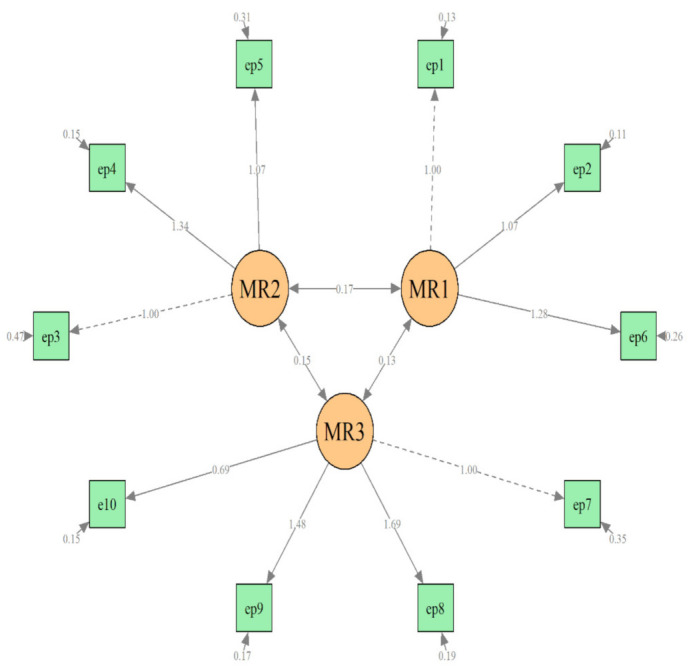
Confirmatory factor analysis of T2 data, 3-factor model (6–8 weeks postpartum).

**Table 1 ijerph-18-06298-t001:** Characteristics of the study participants.

Variable		T1 Total Sample *n* = 584	EPDS Score (SD)	*p* Value	T2 Total Sample *n* = 204	EPDS Score (SD)	*p*-Value
Age (years)	Mean (SD) Range	30.6 (±4.9) 16–44	-	-	30.9 (±4.8) 20–44	-	-
Education	Primary Secondary Tertiary Missing	13 (2.3%) 229 (39.7%) 334 (57.9%) 1 (0.2%)	4.0 (±3.1) 5.1 (±4.2) 5.4 (±4.5)	0.41	1 (0.5%) 62 (31.3%) 135 (68.2%)	4.0 (-) 6.3 (±5.3) 6.2 (±4.8)	0.90
Parity	Primipara Multipara Missing	343 (59.4%) 229 (39.7%) 5 (0.9%)	5.6 (±4.6) 4.7 (±3.8)	0.05	115 (58.1%) 81 (40.9%) 2 (1%)	6.4 (±4.9) 5.8 (±4.9)	0.44
Preterm birth	Yes No Missing	119 (20.6%) 457 (79.2%) 1 (0.2%)	5.8 (±4.1) 5.1 (±4.4)	0.14	40 (20.2%) 158 (79.8%)	5.8 (±4.6) 6.3 (±5.0)	0.50
Type of birth	Vaginal Operative	424 (73.5%) 153 (26.5%)	4.8 (±4.2) 6.5 (±4.4)	0.001	157 (79.3%) 41 (20.7%)	6.2 (±4.8) 6.5 (±4.5)	0.68
Chronic health conditions in anamnesis	Yes No	52 (9.0%) 522 (90.5%) 3 (0.5%)	5.9 (±5.9) 5.2 (±4.2)	0.27	19 (9.6%) 179 (90.4%)	6.4 (±4.9) 6.2 (±5.0)	0.88
Perinatal loss in anamnesis	Yes No Missing	127 (22.0%) 448 (77.6%) 2 (0.3%)	5.2 (±3.8) 5.3 (±4.5)	0.85	41 (20.7%) 157 (79.3%)	6.1 (±4.9) 6.2 (±4.7)	0.90
Support person during labor	Yes No Missing	402 (69.7%) 174 (30.2%) 1 (0.2%)	5.2 (±4.3) 5.3 (±4.6)	0.86	145 (73.2%) 53 (26.8%)	6.3 (±4.6) 6.0 (±5.8)	0.77
History of psychiatric disorder	Yes No	27 (4.7%) 550 (95.3%)	8.8 (±6.1) 5.1 (±4.2)	0.001	7 (3.5%) 191 (96.5%)	11.3 (±4.6) 6.0 (±4.8)	0.05
Risk of major depression	EPDS score ≥ 13 EPDS score ˂ 13	35 (6.1%) 542 (93.9%)	-	-	23 (11.6%) 175 (88.4%)	-	-
Elevated depressive symptoms	EPDS score ≥ 10 EPDS score ˂ 10	96 (16.7%) 481 (83.4%)	-	-	45 (22.7%) 153 (77.3%)	-	-
Zung self-rated depression scale	No signs of depression Minimal signs of depression Moderate signs of depression Severe signs of depression				153 (77.3%) 35 (17.6% 10 (5.1%) -	-	-

**Table 2 ijerph-18-06298-t002:** Cronbach’s α values for the EPDS items.

Item No.	EPDS Cronbach’s α without the Item at T1	EPDS Cronbach’s α without the Item at T2
1. I have been able to laugh and see the funny side of things	0.83	0.87
2. I have looked forward with enjoyment to things	0.83	0.87
3. I have blamed myself unnecessarily when things went wrong	0.83	0.88
4. I have been anxious and worried for no good reason	0.81	0.86
5. I have felt scared or panicky for no very good reason	0.81	0.87
6. Things have been getting on top of me	0.82	0.87
7. I have been so unhappy that I have had difficulty sleeping	0.83	0.88
8. I have felt sad or miserable	0.81	0.86
9. I have been so unhappy that I have been crying	0.82	0.87
10. The thought of harming myself has occurred to me	0.84	0.88
EPDS scale: Cronbach’s α	0.84	0.88
EPDS scale: Spearman Brown coefficient	0.80	0.87

**Table 3 ijerph-18-06298-t003:** Factor matrix for the EPDS in the T1 Slovak sample (exploratory factor analysis with oblimin rotation).

Item No.	Two-Factor Solution		Three-Factor Solution		
	Factor 1	Factor 2	Factor 1	Factor 2	Factor 3
Item 1		0.819		0.755	
Item 2		0.974		0.923	
Item 3	0.520				0.336
Item 4	0.333				0.570
Item 5	0.742				0.997
Item 6	0.420	0.420		0.341	
Item 7	0.641		0.719		
Item 8	0.616	0.363	0.840		
Item 9	0.547		0.754		
Item 10	0.975		0.574		0.473

**Table 4 ijerph-18-06298-t004:** Factor matrix for the EPDS in the T2 Slovak sample (exploratory factor analysis with oblimin rotation).

Item No.	One-Factor Solution	Two-Factor Solution		Three-Factor Solution		
	Factor 1	Factor 1	Factor 2	Factor 1	Factor 2	Factor 3
Item 1	0.888	0.742		0.453	0.372	
Item 2	0.826	0.732			0.901	
Item 3	0.667	0.364	0.368			0.333
Item 4	0.832		0.720			0.634
Item 5	0.690		0.907			0.936
Item 6	0.793	0.439	0.431		0.584	0.369
Item 7	0.662	0.626		0.801		
Item 8	0.832	0.799		0.432	0.440	
Item 9	0.826	1.017		0.827		
Item 10	0.796	0.733		0.752		

## Data Availability

The data presented in this study are available on request from the corresponding author. The data are not publicly available due to ethical and privacy restrictions.

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
