# Peer review of "Factor Structure of the Edinburgh Postnatal Depression Scale in a Sample of Postpartum Slovak Women"

_ijerph, 2021, doi:10.3390/ijerph18126298_

Round 1

Reviewer 1 Report

Škodová et al. examined the application of the Slovak version of the Edinburgh Postnatal Depression Scale (EPDS) on postpartum women. This is an important study showing the practical application of EPDS in Slovak. Although the sample size is not big enough to cover the potential fluctuation of different factors, it’s a good start point. Over all, all statistics were performed pretty properly and the conclusion is convincing. I only have several comments:

  1. For EFA examination for T1 and T2 samples, why the differences are so big (Table 3 and 4)?
  2. It seems CFA only works in T2 data? Why? There are much less samples in T2 samples. So, is there some bias caused this different?
  3. The authors should adjust the format, especially the width of texts and table.

Author Response

# Reviewer 1.:

Škodová et al. examined the application of the Slovak version of the Edinburgh Postnatal Depression Scale (EPDS) on postpartum women. This is an important study showing the practical application of EPDS in Slovak. Although the sample size is not big enough to cover the potential fluctuation of different factors, it’s a good start point. Over all, all statistics were performed pretty properly and the conclusion is convincing. I only have several comments:

Comment 1:

  1. For EFA examination for T1 and T2 samples, why the differences are so big (Table 3 and 4)?

Author’s response:

A possible explanation for these discrepancies might be in the differences between T1 and T2 samples in our study, but also the fact, that T1 data collection was carried out on 2-4th day postpartum- the validity of use EPDS in such a short period after birth might be problematic (please see our response to comment 2 for further details).

Comment 2:

  1. It seems CFA only works in T2 data? Why? There are much less samples in T2 samples. So, is there some bias caused this different?

Author’s response to comment 1 and 2:

Thank you for your comments, this problem is similar for EFA and CFA in our study. The high attrition rate in the T2 sample, which has resulted in a low number of participants in T2, might be attributed to the differences in the data collection process. Baseline T1 data has been collected personally during women’s stay in the hospital (each participant was personally approached by a midwife and invited to participate in the research), which ensured a higher response rate compared to data collection via e-mail in the T2 data collection. We have included a high attrition rate in the T2 sample into the Discussion (paragraph on limitations of the study- lines 412-420).

We have added the non-response bias analysis to our revised manuscript (lines 119-129). Analysis revealed possible non-response bias according to lower education and operative birth in our sample (this was also mentioned in the Discussion part of the revised manuscript (lines 416-420).

Discussion on factors that might contribute to the differences in EFA and CFA between T1 and T2 samples were included in the Discussion (lines 364-374 in the revised manuscript).

Comment 3:

  1. The authors should adjust the format, especially the width of texts and table.

Author’s response:

Thank you for your recommendation, we have adjusted the format of the text and tables in the revised manuscript according to Instruction for Authors.

Reviewer 2 Report

Thank you to the editor and the authors for the opportunity to review this manuscript. The authors examine factor structure and psychometric properties of EPDS in Slovak population. The great value of the study is that second measurement of postpartum depression was made 6-8 weeks after delivery. However I see essential shortcomings of the study methods and procedure that have severe impact on the data interpretation. I summarize the main problem in the following and provide more details in my major comments below.

As the first measurement was taken 2-4 days after delivery it cannot be considered as a measurement of postpartum depression. It is widely recognized that the first onset of postpartum depression is two weeks after birth. Earlier depressive symptoms are either perinatal (with onset during pregnancy) or baby blues. As the study did not take into account the psychiatric history, the history of depression, and the measurement of depression during pregnancy, it is difficult to determine what the T1 results refer to. The three distinguished factors in this period may relate to mood disorders of various etiologies. We could only consider the results of the second measurement (T2 point). Unfortunately, the authors measured depressive symptoms (using ZSDS) only during this period. It is completely unclear why they did not make the same measurement at T1 point. Due to the fact that only 34,9% of participants take part in second measurement, determining who was a participant in the study and to what population we can generalize the results is hard to establish. Authors should carefully define the exclusion and inclusion criteria. This will allow for the appropriate selection of measurement methods. What does "severe psychiatric illness means" and how it was measured? The last major objection to this study concerns the method of measuring symptoms of depression. Although the diagnostic interview is the gold standard, it must be admitted that few EPDS adaptations from other countries take this standard into account. However, choosing ZSDS is totally unclear. The authors do not justify why they chose this questionnaire. In which studies about EPDS the other researchers used this questionnaire?

Major comments:

- line 41: authors stated that no estimation of postpartum depression (PD) were made in Slovak population. However few lines later, they cited theirs own research with depression prevalence made on very similar sample (n=510).

- line 46: the article describes the factor structure of EPDS. So why do the authors in this section described in such detail the factors that predispose to PD? Why is this paragraph important for further analyzes? In this section authors should provide information about factor structure of the EPDS in different samples to justify why is the study of the factor structure important at all?

- Line 100: There is huge attrition rate. Authors verified whether both samples (T1 and T2) differ in terms of the severity of depression. However, there is no information if people who took part in the both part of the research differ from those who refused to participate in T2. How do we know if the higher EDPS scores on T2 are due to the severity of depression or a specific group which decided to participate in the second measurement? If we add to this the fact that we do not know whether the women participated in T1 suffered from depression, we do not know at all who was included to the study population.

- Line 122: "Possible total scores of the scale is ranging from..." I suggest a language proofreading done by a native-speaker.

- Line 142: Why authors used Zung self-rating depression scale made in 1965!? What rationale were behind this choice? When we look to the other adaptation/validation of EPDS, CES-D, HAD-S or clinical interview were utilized. How we can compare this results collected by more than 55 years old questionnaire whit those made by other researchers? Line 151 about ZSDS may be misleading. Article Romera et al. 2008 described the factor structure of ZSDS in patients with depression. The authors themselves point out that this cannot be applied to the general population.

- Line 162: Why depression scale was used only in T2 point? With such a high attrition rate, this is a big limitation of the study. Especially that the authors did not verify the history of depression at all (it is neither an inclusion or exclusion criterion for the study).

- Line 165: " Multiple factor solutions (direct oblimin rotation) were run as previous factor structure models of the EPDS in validation studies differed" - I cannot understand this statement.

- Line 199: how was psychiatric anamnesis conducted? There were no information about it in Measuring instruments section.

- Line 226: if EFA indicates two factors, and parallel analysis three factors, and both models have items in common with this factors - it is worth considering multidimensional factor models: bi-factor or semi-factor models. It seems justifiable to make this calculations, especially that the authors used the laavan package in theirs study.

Final suggestions:

- The authors should consider using the T1 group as a screening group. Participants with an depression score on days 2-4 after delivery should be excluded from further analysis.

- Only group T2 should be included in the examining factor structure of the EPDS.

- It is necessary to analyze whether the people who participated in the study and those who refused differ in studied variables.

- The authors should consider the multidimensional factor models.

Author Response

# Reviewer 2:

Thank you to the editor and the authors for the opportunity to review this manuscript. The authors examine factor structure and psychometric properties of EPDS in Slovak population. The great value of the study is that second measurement of postpartum depression was made 6-8 weeks after delivery. However I see essential shortcomings of the study methods and procedure that have severe impact on the data interpretation. I summarize the main problem in the following and provide more details in my major comments below.

 As the first measurement was taken 2-4 days after delivery it cannot be considered as a measurement of postpartum depression. It is widely recognized that the first onset of postpartum depression is two weeks after birth. Earlier depressive symptoms are either perinatal (with onset during pregnancy) or baby blues. As the study did not take into account the psychiatric history, the history of depression, and the measurement of depression during pregnancy, it is difficult to determine what the T1 results refer to. The three distinguished factors in this period may relate to mood disorders of various etiologies. We could only consider the results of the second measurement (T2 point). Unfortunately, the authors measured depressive symptoms (using ZSDS) only during this period. It is completely unclear why they did not make the same measurement at T1 point. Due to the fact that only 34,9% of participants take part in second measurement, determining who was a participant in the study and to what population we can generalize the results is hard to establish. Authors should carefully define the exclusion and inclusion criteria. This will allow for the appropriate selection of measurement methods. What does "severe psychiatric illness means" and how it was measured? The last major objection to this study concerns the method of measuring symptoms of depression. Although the diagnostic interview is the gold standard, it must be admitted that few EPDS adaptations from other countries take this standard into account. However, choosing ZSDS is totally unclear. The authors do not justify why they chose this questionnaire. In which studies about EPDS the other researchers used this questionnaire?

Author’s response:

Thank you for your comments. We are aware of the methodological limitations of our study. Based on your recommendations we have incorporated several changes, and we believe that these changes have improved the quality of the revised manuscript:

Reviewer 1: Major comments:

Comment 1:

  1. line 41: authors stated that no estimation of postpartum depression (PD) were made in Slovak population. However few lines later, they cited theirs own research with depression prevalence made on very similar sample (n=510).

Author’s response:

We have clarified the intended meaning in the revised manuscript as follows: Only a few studies have estimated the postpartum depression occurrence in the Slovak Republic. A prevalence of 18% was found in a small sample of Slovak postpartum women in the study by Izakova (2013), similarly, 25% prevalence was reported in a study by Banovcinova, Skodova, & Jakubcikova, (2018). (lines 40-43 in the revised manuscript)

Comment 2:

  1. line 46: the article describes the factor structure of EPDS. So why do the authors in this section described in such detail the factors that predispose to PD? Why is this paragraph important for further analyzes? In this section authors should provide information about factor structure of the EPDS in different samples to justify why is the study of the factor structure important at all?

Author’s response:

Based on this comment, we have added the short overview of the EPDS factor structure in recent previous studies into the Introduction part of the manuscript (78-84 lines in the revision).

Based on comments from Reviewer 3, who suggested adding more risk factors of postpartum depression in the Introduction, we have decided to keep the paragraph on risk factors in the revised manuscript. Our intention in the first three paragraphs of the Introduction was to provide an overview of postpartum depression as the key concept in our study.

Comment 3:

  1. Line 100: There is huge attrition rate. Authors verified whether both samples (T1 and T2) differ in terms of the severity of depression. However, there is no information if people who took part in the both part of the research differ from those who refused to participate in T2. How do we know if the higher EDPS scores on T2 are due to the severity of depression or a specific group which decided to participate in the second measurement? If we add to this the fact that we do not know whether the women participated in T1 suffered from depression, we do not know at all who was included to the study population.

Author’s response:

Thank you for this useful comment. We have added the non-response analysis to our revised manuscript (lines 119-129), which revealed possible non-response bias according to lower education and operative birth in our sample. No significant differences in t2 response rate regarding EPDS positive scores in T1 have been found. We have added non-response bias also to the Discussion part of the revised manuscript (lines 413-418).

 Comment 4:

  1. Line 122: "Possible total scores of the scale is ranging from..." I suggest a language proofreading done by a native-speaker.

Author’s response:

Proofreading by professional language services has been performed in the revised manuscript.

 Comment 5:

  1. Line 142: Why authors used Zung self-rating depression scale made in 1965!? What rationale were behind this choice? When we look to the other adaptation/validation of EPDS, CES-D, HAD-S or clinical interview were utilized. How we can compare this results collected by more than 55 years old questionnaire whit those made by other researchers? Line 151 about ZSDS may be misleading. Article Romera et al. 2008 described the factor structure of ZSDS in patients with depression. The authors themselves point out that this cannot be applied to the general population.

Author’s response:

We realize that using ZSDS as the measuring scale of depressive symptoms in our study is a methodological limitation, we have added this to the discussion section of the revised manuscript (420-426 lines in revision- Discussion) and we also added an explanation for choosing this measure in our study to the Methods section (lines 162-165)

Comment 6:

  1. Line 162: Why depression scale was used only in T2 point? With such a high attrition rate, this is a big limitation of the study. Especially that the authors did not verify the history of depression at all (it is neither an inclusion or exclusion criterion for the study).

Author’s response:

Thank you for this comment, in the revised manuscript, we have clarified the reasons for the high attrition rate: we assumed that this might be explained by the differences in the data collection process. Baseline T1 data has been collected personally during women's stay in the hospital (each participant was personally approached by a midwife and invited to participate in the research), which ensured a higher response rate compared to data collection via e-mail in the T2 data collection. (lines 114-118). We realized that use of the Zung depression scale only in T2 is a methodological problem, this was addressed in the Discussion part of the revised manuscript (420-426).

 Comment 7:

  1. Line 165: " Multiple factor solutions (direct oblimin rotation) were run as previous factor structure models of the EPDS in validation studies differed" - I cannot understand this statement.

Author’s response:

We have clarified this in the revised manuscript as follows: Multiple factor solutions (direct oblimin rotation) were run in our analysis.

 Comment 8:

  1. Line 199: how was psychiatric anamnesis conducted? There were no information about it in Measuring instruments section.

Author’s response:

  1. Psychiatric anamnesis of the participants was measured by 2 self-reported questions included in the sociodemographic and perinatal questionnaire, questions were focused on the history of depressive diagnosis or other psychiatric illness in personal history. This information was added to the Methods section- (lines 166-172 in revision).

Comment 9:

  1. Line 226: if EFA indicates two factors, and parallel analysis three factors, and both models have items in common with this factors - it is worth considering multidimensional factor models: bi-factor or semi-factor models. It seems justifiable to make this calculations, especially that the authors used the laavan package in theirs study.

 Author’s response:

In our validation study, we have focused on basic procedures within the EFA and CFA, unfortunately, we were not able to include suggested analysis in our revised manuscript.

Reviewer 2: Final suggestions:

  1. The authors should consider using the T1 group as a screening group. Participants with an depression score on days 2-4 after delivery should be excluded from further analysis. Only group T2 should be included in the examining factor structure of the EPDS.

 Author’s response:

We have considered excluding analysis of the factor structure in the T1 sample from our study as suggested, however, we believe that providing details on EFA and CFA analysis in both samples will contribute to more complex information about the results of our study.

We have included a discussion in our revised manuscript, based on your recommendations and concerns regarding the appropriateness of using both T1 and T2 samples. Following issues has been addressed in the Discussion section of the revised manuscript: the fact that validity of use EPDS in such a short period after birth might be problematic, the differences in EFA and CFA results between T1 and T2 samples in our study, and high attrition rate in T2 sample (please see lines 364-374 in the revised manuscript).

  1. It is necessary to analyze whether the people who participated in the study and those who refused differ in studied variables.

 Author’s response:

Thank you for this useful recommendation, this has been addressed in the Major comments, please see our response on Comment 3.

  1. The authors should consider the multidimensional factor models.

Author’s response:

This has been addressed in the Major comments, please see our response on Comment 9.

Reviewer 3 Report

Attached

Author Response

# Reviewer 3:

The presented study tackles an issue of Factor Strudcture of the Edinburg Postnatal Depression Scale in a Sample of Slovak Postpartum Women. The study was conducted reliably with appropriate selection of tests. Overall, I think that this article should be published, however after reedition the text according to Instructions for Authors. Some issues require complementary information:

 Comment 1-3:

  1. I suggest capitalizing the title.
  2. I suggest changing the Citation Style – See the Instructions for Authors (Zotero id:multidisciplinary-digital-publishing-institutem, Endnote Non-superscripted Number)
  3. I suggest including the tables in the text- See the Instructions for Authors

Author’s response:

Thank you for these comments regarding formatting and referencing style, these issues have been changed in the revised manuscript according to the Instruction for Authors and IJERPH Template provided by the editorial office.

 Comment 4:

  1. I suggest including in the introduction more risk factors of pospartum depression eg. Lyuvenova et al. Depression prevalence based on the Edinburgh Postnatal Depression Scale compared to Structured Clinical Interview for DSM DIsorders classification: Systematic review and individual participant data meta-analysis · Zaręba et al. Peripartum Predictors of the Risk of Postpartum Depressive Disorder: Results of a Case-Control Study.

Author’s response:

Thank you for your recommendations, we have included a study by Zareba et al. in the Introduction section in the paragraph on risk factors of postpartum depression (line 62 in the revised manuscript), and a study by Lyuebenova et al. has been included in the Discussion section in the paragraph on postpartum depression prevalence in different countries (line 392).

 Comment 5:

  1. I suggest including why the cut-off point for EDPSC Score was so high (>13)

Author’s response:

In our study, we have employed both cut-off points for EPDS scores (>13) and (>10), which are most frequently used in validation studies of the EPDS. Cut-off point (>13) have been recommended for distinguishing clinically relevant depression symptom by authors of the scale (EPDS manual- 2nd edition, 2014; Cox, J.: Thirty years with the Edinburgh Postnatal Depression Scale, 2019), this has been mentioned in the Methods section (lines 141-146) in the revised manuscript.

 Comment 6:

  1. I suggest using only one author in the text – it’s more clear ( e.g.: verse 27 „Shore et al.”)

Author’s response:

             This has been corrected in the whole manuscript

 Comment 7:

  1. Verse 46-50 I suggest reediting that sentence- it’s to long.

Author’s response:

Thank you for this comment, sentence has been shortened as follows: “According to a systematic review by Hutchens, & Kearney, (2020), the risk factors for postnatal depression are high life stress, lack of social support, current or past abuse, prenatal depression, marital or partner dissatisfaction, and prenatal depression” (lines 46-49 in the revised manuscript)

 Comment 8:

  1. Verse 74 I suggest deleting „has been”- it’s duplicated

Author’s response:

This error has been deleted.

 Comment 9:

  1. Verse 102-111 I suggest including that phrases in the Results

Author’s response:

Description of the basic sociodemographic and anamnestic factors was included in the Results section of the revised manuscript as suggested (lines 212-222)  in the revised manuscript). Based on other comments from reviewers we have rewrite the section on data collection to provide more details on sampling procedures (lines 92-109).

 Comment 10:

  1. Verse 155-156 I suggest including the year of production of the statistical softwere package

Author’s response:

This has been added to the paragraph on statistical analysis (line 175)

 Comment 11:

  1. I suggest incuding Author contribution, Funding etc. According to Instructions for Author

Author’s response:

Thank you for your recommendation, we have included the Authors contribution as well as Funding, Data availability, and Conflict of interest to the revised manuscript (lines 448-465).

Reviewer 4 Report

The topic of this manuscript is relevant and very important in the postpartum clinical care in Slovak. In general, the report provide appropriate information about the process of validation, but please address:

  1. The low response rate at T2. It should be explained in the abstract and limitations.
  2. Convergent validity was performed only at T2 and it need to be justified. In the same way, the use of Zung Scale is not sufficiently justified.
  3. Reliability an validity measures were appropriate but it could be necessary to explain why you do not offer information about sensitivity in the validation process performing the calculation of the interclass correlation coefficient about the differences between the measurements for each subject.
  4. Table 5 it could be eliminated.

Author Response

# Reviewer 4:

The topic of this manuscript is relevant and very important in the postpartum clinical care in Slovak. In general, the report provide appropriate information about the process of validation, but please address:

Comment 1:

  1. The low response rate at T2. It should be explained in the abstract and limitations.

Author’s response:

Thank you for this comment. We have added our explanation of the high attrition rate in the Methods part of the manuscript (lines 114-118), limitations (lines 413-420), and abstract (lines 10-13). Baseline T1 data has been collected personally during women’s stay in the hospital (each participant was personally approached by a midwife and invited to participate in the research), which ensured a higher response rate compared to data collection via e-mail in the T2 data collection.

Comment 2:

  1. Convergent validity was performed only at T2 and it need to be justified. In the same way, the use of Zung Scale is not sufficiently justified.

Author’s response:

We are aware of the fact, that using ZSDS as the measuring scale of depressive symptoms in our study, as well the fact, that ZSDS was used only in T2 point, are methodological limitations. We have added this to the discussion section of the revised manuscript (420-426 lines in revision- Discussion), and we also added an explanation for choosing this measure in our study to the Methods section (lines 162-165).

 Comment 3:

  1. Reliability an validity measures were appropriate but it could be necessary to explain why you do not offer information about sensitivity in the validation process performing the calculation of the interclass correlation coefficient about the differences between the measurements for each subject.

Author’s response:

Thank you for this useful recommendation, we have added the results of the interclass correlation coefficients into the revised manuscript (lines 183-185, and 256-258).

Comment 4:

  1. Table 5 it could be eliminated.

Author’s response:

Thank you for this comment, we have deleted Tabel 5 from the revised manuscript, and a short paragraph on the factor structure of the EPDS in previous studies has been added to the Introduction section instead (lines 78-85).

Round 2

Reviewer 2 Report

I read carefully the revised manuscript and the responses to the review. Unfortunately, there are still essential shortcomings of the study that need to be addressed. The most important of these is the discrepancy regarding the number of respondents and why people diagnosed with depression and mental disorders (by self-report question) were not excluded from the study. The responses to the reviews revealed that the participants were asked about it during the study, and this remark could not be taken into account in first review.

Line 147 - The high attrition rate in T2 might be explained.... - this should be moved to the study limitation section. In this paragraph authors should provide only information about group differences (from line 156).

Line 167 - did you used t test to compare EPDS scores among both groups?

Line 220 - "Psychiatric anamnesis of the participants was measured by 2 self-reported questions focused on a history of depressive diagnosis and other psychiatric illness..." First of all, what do you mean by "depressive diagnosis" - do you mean "history of depression disorder"? Second, how you diagnosed depression by this questions? You ask for previous onset of depression, actual or both? This question refers to self-diagnosis or psychiatric diagnosis? And last but not least, author's shouldn't name the 2 self-report questions as a psychiatric anamnesis.

Line 290 - "...were only found regarding psychiatric anamnesis (women with positive psychiatric anamnesis scored significantly higher..." - may you explain who were included to this group? Those who reported depression and psychosis, only those who reported depression? How you merged this two self-reported questions into one group?

Line 220 - Why were participants with a history of mental illness not excluded from the study? You had 27 participants who reported such history. You should excluded it from the study and make the calculations again.

Line 135/line 271 and Table 1 - How many participants do you have in your study? According to the Participant section N=584. So why you have only N=577 EPDS scores in Table1? This is your main variable, so you shouldn't treat it like missing data. The seme case is in reporting T2 characteristics. We can read that you have 204 participants in this group. However you have provided only 198 scores of EPDS and Zung scale.

Table 1 - please move the scores to exact rows in "Risk of major depression" and "Elevated depressive symptoms"

Table 1 - All percentages in the table are misleading. For example - you have 581 women who answered for Perinatal loss (it means that n=3 didn't answer for this question). However, all percentages summarize to 100% (22%+78%). This problem occurs in all reported variables : i.e. Preterm birth and Psychiatric anamnesis (n=583), Parity (n=578). I could understand that in all research we have missing data, but you have to include it in the table (especially when you provided group's percentages).

Line 580 - Authors should add to the limitations sections: no history of depression diagnosis (other than one self-report question);

Line 421 - In author response for comment 9: "...unfortunately, we were not able to include suggested analysis in our revised manuscript". Adding the bi-factor and second-order factor models to the analysis would increase the value of the analyzed data. Performing this analysis in R is very simple:

# 1 second-order factor analysis (You need f1, f2 and f3 (three factors where q01 is an item name) and f4 as a second order factor (f4 =~ 1*f1 + 1*f2 + 1*f3):

#second order three factor solution, marker method

m5a <- 'f1 =~ q01+ q03 + q04 + q05 + q08

        f2 =~ q06 + q07 

        f3 =~ 1*f1 + 1*f2

        f3 ~~ f3'

secondorder <- cfa(m5a, data=dat)

summary(secondorder,fit.measures=TRUE,standardized=TRUE)

#2 bi-factor model (general.factor = all items).

m.model.2 <- '

general.factor  =~ x1 + x2 + x3 + x4 + x5 + x6 + x7 + x8 + x9 + x10 + x11 + x12

factor1         =~ x1 + x2 + x3 + x4 factor2         =~ x5 + x6 + x7 + x8 factor3         =~ x9 + x10 + x11 + x12 general.factor  ~~ 0*factor1 general.factor  ~~ 0*factor2 general.factor  ~~ 0*factor3 factor1         ~~ 0*factor2 factor1         ~~ 0*factor3 factor2         ~~ 0*factor3   m.modelfit.2 <- cfa(m.model.2, data = bifac, std.lv = TRUE, information="observed") summary(m.modelfit.2, fit.measures = TRUE, standardized = TRUE)    

Author Response

# Reviewer 2:

I read carefully the revised manuscript and the responses to the review. Unfortunately, there are still essential shortcomings of the study that need to be addressed. The most important of these is the discrepancy regarding the number of respondents and why people diagnosed with depression and mental disorders (by self-report question) were not excluded from the study. The responses to the reviews revealed that the participants were asked about it during the study, and this remark could not be taken into account in first review.

Comment 1:

Line 147 - The high attrition rate in T2 might be explained.... - this should be moved to the study limitation section. In this paragraph, authors should provide only information about group differences (from line 156).

Author’s response:

The explanation of the high attrition rate was moved to the Discussion- paragraph on study limitations (lines 406-409 in second revision).

 Comment 2:

Line 167 - did you used t test to compare EPDS scores among both groups?

Author’s response

This analysis has been added to the revised manuscript (lines 126-128): Total EPDS scores in response and non-response group were compared using Student independent samples t-test, and no significant differences were found (t= 1.90, p>0.05).

Comment 3:

Line 220 - "Psychiatric anamnesis of the participants was measured by 2 self-reported questions focused on a history of depressive diagnosis and other psychiatric illness..." First of all, what do you mean by "depressive diagnosis" - do you mean "history of depression disorder"? Second, how you diagnosed depression by this questions? You ask for previous onset of depression, actual or both? This question refers to self-diagnosis or psychiatric diagnosis? And last but not least, author's shouldn't name the 2 self-report questions as a psychiatric anamnesis.

Author’s response:

This has been amended in the revised manuscript as follows: Previous or actual onset of psychiatric disorders of the participants was measured by 2 self-reported questions focused on a history of depression or other psychiatric illness before or during pregnancy. (lines 169-171)

 Comment 4:

Line 290 - "...were only found regarding psychiatric anamnesis (women with positive psychiatric anamnesis scored significantly higher..." - may you explain who were included to this group? Those who reported depression and psychosis, only those who reported depression? How you merged this two self-reported questions into one group?

Author’s response:

Due to a small number of women who reported history of depression or other psychiatric disorder before or during pregnancy, we compared two groups of participants: the first group included those who didn’t report any history of psychiatric illness, and the second group included those participants who reported either depression or other psychiatric illness in their history. This has been revised also in the manuscript (lines 225-227)

 Comment 5:

Line 220 - Why were participants with a history of mental illness not excluded from the study? You had 27 participants who reported such history. You should excluded it from the study and make the calculations again.

Author’s response:

We have carefully considered this requirement; however, we do not see a rationale for the exclusion of women with a history of mental illness from the study sample. To our knowledge, this is not a standard procedure in the validation studies of the EPDS, and we did not find such exclusion criteria in other EPDS validation studies in UK, Hungary, Serbia or the US (references 16,17,18,20, 30, 31, 32, 33 in the revised manuscript). Moreover, exclusion of women with a history of mental health problems have resulted in the significantly worse values of the Kaiser–Meyer–Olkin measure of sampling adequacy for the T2 sample (0.67 compared to the original value of 0.82)- optimal value for factor analysis is higher than 0.80.

Comment 6:

Line 135/line 271 and Table 1 - How many participants do you have in your study? According to the Participant section N=584. So why you have only N=577 EPDS scores in Table1? This is your main variable, so you shouldn't treat it like missing data. The seme case is in reporting T2 characteristics. We can read that you have 204 participants in this group. However you have provided only 198 scores of EPDS and Zung scale.

Author’s response:

The following sections of the manuscript have been updated: Table 1, the Methods section (213-220 lines in the revised manuscript), and Abstract (lines 11-13). Together 577 respondents in the T1 sample and 198 respondents in the T2 sample has been reported in the revised manuscript

Comment 7:

Table 1 - please move the scores to exact rows in "Risk of major depression" and "Elevated depressive symptoms"

Author’s response:

Due to the fact, that different cut-off scores of the EPDS scale are used in other research studies, we believe it is meaningful to keep both data with a 10-point cut-off score, and a 13-point cut-off score in Table 1.

Comment 8:

Table 1 - All percentages in the table are misleading. For example - you have 581 women who answered for Perinatal loss (it means that n=3 didn't answer for this question). However, all percentages summarize to 100% (22%+78%). This problem occurs in all reported variables : i.e. Preterm birth and Psychiatric anamnesis (n=583), Parity (n=578). I could understand that in all research we have missing data, but you have to include it in the table (especially when you provided group's percentages).

Author’s response:

Table 1 has been updated- missing data has been included, and percentages have been calculated again.

Comment 9:

Line 580 - Authors should add to the limitations sections: no history of depression diagnosis (other than one self-report question);

Author’s response:

This has been added to the limitations section (lines 418-419 in the revised manuscript).

Comment 10:

Line 421 - In author response for comment 9: "...unfortunately, we were not able to include suggested analysis in our revised manuscript". Adding the bi-factor and second-order factor models to the analysis would increase the value of the analyzed data. Performing this analysis in R is very simple:

Author’s response:

We have performed the suggested analysis, however, the RMSEA values and both CFI and TLI values of the models have been shown as unsatisfactory, which was the reason for not including those models into the revised manuscript.

Please find below the description of the models:

Bi-factor model: RMSEA = 0.165, CFI=0.80, TLI= 0.64

Second-order model: RMSEA =0.092, CFI= 0.91, TLI=  0.89

Reviewer 3 Report

I have no other comments.

Author Response

Thank you.